# Metabolic Dysfunction-Associated Steatotic Liver Disease in Japan: Prevalence Trends and Clinical Background in the 10 Years before the Coronavirus Disease 2019 Pandemic

**DOI:** 10.3390/medicina60081330

**Published:** 2024-08-16

**Authors:** Akira Sato, Yumiko Oomori, Rika Nakano, Tomokazu Matsuura

**Affiliations:** 1Department of Health Management, St. Marianna University Yokohama Seibu Hospital, 1197-1 Yasashicho Asahi-ku, Yokohama 241-0811, Kanagawa, Japan; 2Medical Department, Sasaki Foundation Shonan Health Examination Center, 10-4 Takaracho, Hiratsuka 254-0034, Kanagawa, Japan; 3Department of Clinical Examination, Sasaki Foundation Shonan Health Examination Center, 10-4 Takaracho, Hiratsuka 254-0034, Kanagawa, Japan; 4Department of Radiology, Sasaki Foundation Shonan Health Examination Center, 10-4 Takaracho, Hiratsuka 254-0034, Kanagawa, Japan; rnakano@po.kyoundo.jp

**Keywords:** metabolic dysfunction-associated steatotic liver disease, epidemiology, dietary nutrient, metabolic syndrome, lifestyle-related disease, prediabetes, dietary fat

## Abstract

*Background and Objectives:* The trends in metabolic dysfunction-associated steatotic liver disease (MASLD) and related metabolic dysfunctions in Japan are unknown. Thus, we aimed to clarify these trends before the novel coronavirus disease 2019 pandemic in Japan. *Materials and Methods:* We included Japanese individuals aged 25–79 years who underwent health examinations at our center. We analyzed anthropometry, lifestyle-related disease, and nutritional intake in relation to MASLD trends from 2010–2019. *Results:* The prevalence of MASLD increased in all ages and body mass index (BMI) classes, reaching 30.3% in males and 16.1% in females, with MASLD accounting for 75% of steatotic liver cases and more than half of all type 2 diabetes mellitus (T2DM) and high waist circumference (HWC) cases. The increase in the prevalence of MASLD was thought to be largely attributable to an increase in that of the incidence of steatotic liver itself, and there was no increase in the prevalence of other factors, such as overweight, T2DM, hypertension, and dyslipidemia. The prevalence of glucose metabolic disorders (GMDs) and hypertension decreased. National nutritional data showed an increase in energy intake, total fat, saturated fatty acids, monounsaturated fatty acids, and polyunsaturated fatty acids, which correlated with a decrease in GMDs. Salt intake also decreased, which correlated with hypertension. The MASLD group had a higher prevalence of all related metabolic factors than the non-MASLD group, especially HWC, T2DM, and hyperlipidemia. *Conclusions:* The prevalence of MASLD increased with that of steatotic liver, regardless of age or BMI. A relationship between increased dietary fat, increased steatotic liver, and decreased GMDs was suggested.

## 1. Introduction

Non-alcoholic fatty liver disease (NAFLD) is defined as a chronic liver disease characterized by excessive fat accumulation in the liver without another obvious cause (such as excessive alcohol consumption, viral infections, autoimmune hepatic diseases, or hepatotoxic medications). Histologically, NAFLD is divided into two categories: non-alcoholic fatty liver (NAFL) and non-alcoholic steatohepatitis (NASH) [1]. The former is defined as fatty liver without evidence of hepatocellular damage or liver fibrosis, whereas NASH is defined as fatty liver with hepatocellular damage (ballooning). NAFL has a low risk of developing liver disease, whereas NASH often progresses to cirrhosis and hepatocellular carcinoma.

NAFLD is often associated with metabolic abnormalities, including obesity, insulin resistance, type 2 diabetes mellitus (T2DM), hypertension, and dyslipidemia, which influence the progression of liver pathology [2,3,4]. Cardiovascular disease (CVD) is the leading cause of death in NAFLD, followed by extrahepatic cancer and liver disease [5]; T2DM is associated with all-cause mortality and liver-related events in NAFLD [2], hypertension increases total adiposity and cardiac mortality; and dyslipidemia is directly related to CVD and promotes liver fibrosis [3,4]. Therefore, it is important to identify patients with NAFLD and these metabolic abnormalities, including obesity, to adequately determine NAFLD risk. In addition, the term NAFLD is a negative diagnosis that excludes secondary causes of liver disease (such as alcohol and viruses), and the term fatty liver is associated with some stigma.

Against this background, an International Consensus Group changed the term “fatty liver disease” to “steatotic liver disease” (SLD), and proposed the term “metabolic dysfunction-associated SLD (MASLD)” as an alternative to NAFLD [6,7]. As MASLD is an NAFLD associated with one or more metabolic disorders, this revision is significant because it helps to clarify the etiology of the disease and identify those at high risk. Although there are many cross-sectional epidemiologic data on metabolic abnormalities associated with NAFLD [8,9], there are few longitudinal epidemiologic studies [10,11].

In this study, we sought to examine trends in MASLD and related metabolic dysfunctions before the onset of the novel coronavirus disease 2019 (COVID-19) pandemic in Japan. We found that the prevalence of MASLD increased with that of steatotic liver, regardless of age or body mass index (BMI). In addition, pre-diabetes decreased in related metabolic disorders, suggesting that dietary nutrition may play a role in this process.

## 2. Materials and Methods

### 2.1. Study Population and Design

Japanese individuals aged 25–79 years who underwent physical examinations, physiological examinations, abdominal ultrasonography (US), and blood screening examinations at the Sasaki Foundation Shonan Health Screening Center in the Kanto region during the fiscal years (FYs) 2010–2019 (April to March of the following year) were included in this study. Participants were included if they met the following criteria: (1) available BMI data, (2) available fasting blood glucose or glycated hemoglobin (HbA1c) data, and (3) known alcohol intake. Of the 48,225 examinees, 31,378 (16,238 males and 15,140 females) met the inclusion criteria. The characteristics of the subjects are shown in Table 1.

The study protocol was approved by the Human Ethics Review Committee of the Sasaki Foundation (Tokyo, Japan) (certification no. 19728). Informed consent was obtained on an opt-out basis.

MASLD was diagnosed according to the NAFLD Nomenclature Consensus Group criteria [6]. High waist circumference (HWC) was defined as ≥85 cm for males and ≥90 cm for females, according to the Japanese metabolic syndrome diagnostic criteria [12]. Age was classified into three categories according to the criteria of the Ministry of Health, Labour and Welfare: 25–44 years, mature age; 45–64 years, middle age; and ≥65 years, old age. BMI was classified into the following categories according to the recommendation of the Western Pacific Region of the World Health Organization criteria that pertain to obesity: <23.0 kg/m^2^ as normal, ≥23.0 kg/m^2^ as overweight, 23–24.9 kg/m^2^ as pre-obese (pre-Ob), 25–29.9 kg/m^2^ as obese-I (Ob-I), and ≥30 kg/m^2^ as obese-II (Ob-II).

All lifestyle-related disease (LRDs: hypertension, glucose metabolism disorders (GMDs), and dyslipidemia) diagnoses were defined according to the International Consensus Group criteria for metabolic dysfunction [6] and were defined as those that were currently being treated with medications or which met the following criteria: hypertension, defined as systolic blood pressure > 130 mmHg and/or diastolic blood pressure > 85 mmHg; GMDs, defined as fasting plasma glucose (FPG) > 100 mg/dL or HbA1c > 5.7%; type 2 diabetes mellitus (DM) (T2DM), defined as FPG > 126 mg/dL or HbA1c > 6.5%; prediabetes, defined as a FPG ≥ 100 mg/dL and <126 mg/dL or HbA1c ≥ 5.7% and <6.5% [13]; and dyslipidemia, defined as triglyceride levels of ≥150 mg/dL or a high-density lipoprotein cholesterol level of <40 mg/dL for males and <50 mg/dL for females. HOMA-IR, one of the GMD diagnostic parameters, was not included in the evaluation criteria because blood insulin levels were not measured.

### 2.2. Physical Examinations and Serum Biochemistry Analyses

Body weight and height were obtained for both sets of participants, and BMI was calculated. Skilled nurses measured waist circumference at the navel while the participants were in a standing position. Venous blood samples were obtained from all participants from 8–10 a.m. following a 12 h overnight fast.

### 2.3. Protocol for Abdominal US and Definition of Fatty Liver

All participants underwent abdominal US to assess steatotic livers. The liver parenchyma of all patients was examined using a conventional convex array transducer. The presence of steatotic change was defined as increased echogenicity of the liver parenchyma compared with the renal parenchyma (bright liver and liver-kidney contrast), deep US attenuation in the right lobe of the liver (deep attenuation), and/or poor visualization of the hepatic vein (vascular blurring); the first two were considered definitive criteria, while the latter two were considered necessary. The US systems used were the Aloka SSDα5, Aloka Pro Soundα7, Aloka Pro Soundα7, and Aloka ARIETTA E50 (Hitachi Aloka Medical, Tokyo, Japan), with 3.5 MHz convex array transducers. Experienced sonographers trained by gastroenterologists performed all examinations. The technical parameters were adjusted for each subject using a standard US protocol. A board-certified gastroenterologist from the Japanese Society of Gastroenterology reviewed the images and diagnosed steatotic liver without referring to any other personal data of the participant.

### 2.4. Dietary Data Source

We used dietary data from the National Health and Nutritional Survey (NHNS) of Japan conducted by the National Institute of Health and Nutrition [14] The NHNS is a cross-sectional household interview and examination survey conducted annually since 1945 by local health centers under the supervision of the Ministry of Health, Labour and Welfare of Japan [15]. Details of the survey design have been described elsewhere [16]. This survey is conducted annually throughout Japan in November.

The number of households participating in the survey between 2010 and 2019 ranged from 2836 (2019) to 12,750 (2012), with a response rate of approximately 50%. A total of 118,840 individuals participated in the survey between 2010 and 2019. This study used information from 98,403 individuals (4927–26,726/year; average 9840/year) aged ≥20 years in the NHNS survey to match the age distribution of the study population. Specifically, the number of people in each age group in the NHNS was converted to the number in the study population and the mean dietary nutrient level for each year was recalculated.

### 2.5. Statistical Analysis

Trend analyses of the prevalence of MASLD, LRDs, obesity, HWC, dietary nutrient intake, and MASLD were performed using Spearman’s rank correlation, and comparisons between MASLD and non-MASLD were performed by the Chi-square and Student’s *t*-test with Stat Flex, version 7 (Artec, Osaka, Japan). A *p*-value <0.05 was considered statistically significant.

In addition, scatter plots were generated for the correlation of dietary nutrient intake and GMDs using Microsoft Excel, version 2403 (Microsoft Corporation, Redmond, WA, USA). The coefficients of determination (R^2^) were calculated, and the coefficients were interpreted according to Schober et al. [17].

## 3. Results

### 3.1. Trends in MASLD and Steatotic Liver

The prevalence of MASLD increased from 21.8% to 30.3% for males and from 10.4% to 16.1% for females over the 10-year period ending in fiscal year (FY) 2018–2019 (Figure 1). This increase was linked to the prevalence of steatotic liver itself and of NAFLD (ρ = 1.0, *p* < 0.05 for both sexes), with the most recent MASLD overlap rates of 71.8% for steatotic liver and 95.3% for NAFLD in males and 86.8% and 94.1%, respectively, in females.

### 3.2. Trends in MASLD by Age and BMI

The prevalence of MASLD by age group increased in all age groups in males, especially in mature (from 21.9% to 29.1%) and middle age (from 22.9% to 32.5%) (*p* < 0.05) (Figure 2a,b). The prevalence also increased in females of all age groups (mature: 5.3% to 7.2%, middle: 13.1% to 19.3%, and old: 13.2% to 25.1%; *p* < 0.05).

The MASLD prevalence according to BMI increased in all BMI groups (*p* < 0.05), except for the Ob-II group in males, especially in the pre-Ob (18.8% to 29.5%) and Ob-I groups (40.9% to 54.3%) (Figure 2c,d). In females, there was a significant increase in the pre-Ob (from 14.1% to 27.1%), Ob-I (from 34.3% to 47.5%), and Ob-II (from 62.2% to 81.4%) groups (*p* < 0.05).

### 3.3. Trends in MASLD Factors

Regarding physical factors, the prevalence of steatotic liver increased in both males and females (from 29.4% to 43.4% in males and from 11.3% to 18.5% in females; *p* < 0.005) (Figure 3). However, there was no significant change in the prevalence of overweight (range: males, 57.2–58.9% and females, 31.0–31.5%) and HWC (range: males, 38.9–42.4% and females, 8.7–10.5%).

Regarding metabolism-related factors, both males (62.0% to 55.7%) (*p* = 0.0381) and females (45.0% to 31.4%) (*p* = 0.0042) had a decreased prevalence of GMDs, and the proportion of participants with prediabetes decreased (males: 52.1% to 46.0%, *p* = 0.0323 and females: 41.8% to 31.4%, *p* = 0.0037). The prevalence of hypertension also showed a decreasing trend from 47.0% to 37.0% (*p* = 0.0601) in males, and a significant decrease from 26.5% to 19.8% (*p* = 0.0102) in females.

### 3.4. Nutritional Intake in the NHNS and Their Relationship to GMDs and Hypertension

Trends in nutrient intake according to the NHNS showed an increase in energy (EN) intake for both sexes (Table 2); a decreasing but not significant trend in carbohydrate (CHO) intake; and an increase in total fat, saturated fatty acid (SFA), monounsaturated fatty acid (MUFA), and polyunsaturated fatty acid (PUFA) intake for both sexes. Salt intake tended to decrease in both sexes, but was only significant in males.

EN intake showed an increasing trend but a strong inverse correlation with the prevalence of GMDs in males (R^2^ = 0.5146) and GMDs and prediabetes in females (R^2^ = 0.806 and R^2^ = 0.7925, respectively), and a moderate inverse correlation with the prevalence of prediabetes in males (R^2^ = 0.4633) (Figure 4a,b). CHO levels were strongly inversely correlated with the prevalence of GMDs and prediabetes in females (R^2^ = 0.7237 and R^2^ = 0.7238, respectively) (Figure 4d). Total fat showed a moderate inverse correlation with the prevalence of GMDs and prediabetes in males (R^2^ = 0.5848 and R^2^ = 0.5987, respectively) and a strong inverse correlation in females (R^2^ = 0.918 and R^2^ = 0.9109, respectively) (Figure 4e,f). SFA and MUFA showed moderate correlations in males (SFA: R^2^ = 0.4305 and R^2^ = 0.446 and MUFA: R^2^ = 0.4444 and R^2^ = 0.4583, respectively) (Figure 4g,i) and strong inverse correlations with the prevalence of GMDs and prediabetes in females (SFA: R^2^ = 0.7714 and R^2^ = 0.7615 and MUFA: R^2^ = 0.7346 and R^2^ = 0.7244, respectively) (Figure 4h,j). PUFA showed weak correlations in males (R^2^ = 0.3561 and R^2^ = 0.3765, respectively) and strong inverse correlations with the prevalence of GMDs and prediabetes in females (R^2^ = 0.7323 and R^2^ = 0.7194, respectively) (Figure 4k,l).

Salt intake was significantly reduced in males but weakly correlated with the prevalence of hypertension (R^2^ = 0.342). However, there was a moderate correlation with a reduced prevalence of hypertension in females (R^2^ = 0.483), where the reduction in salt intake was not significant (Figure 5a,b).

### 3.5. Clinical Features of MASLD

The MASLD group had a higher percentage of males (65.5%) (Table 3), mean age (52 years), and prevalence of overweight and HWC (85.2% and 60.9%, respectively) than did the non-MASLD group. Hypertension, dyslipidemia, T2DM, and prediabetes were present in 44.5%, 51.2%, 14.7%, and 53.2% of participants, respectively. In the MASLD group, 46.6% of patients had dyslipidemia and 53.4% had T2DM. BMI and various laboratory values other than platelet counts deviated more frequently from the reference values in patients with MASLD than in those without.

## 4. Discussion

This study revealed the following points. First, the prevalence of MASLD increased in almost all age and BMI groups of both sexes before the COVID-19 pandemic, which was associated with an increase in the prevalence of steatotic liver. Second, both males and females with MASLD-related LRDs exhibited a reduced prevalence of prediabetes. This was inversely correlated with the total fat and various fatty acid (FA) intakes in the national data. Third, MASLD was more prevalent in males than in individuals without MASLD, and individuals with MASLD were older and exhibited a higher prevalence of overweight, HWC, and LRDs than individuals without MASLD.

The prevalence of MASLD increased in almost all age and BMI groups as the prevalence of steatotic liver increased. In other words, MASLD is composed of multiple physical and metabolic factors, but no factors other than steatotic liver were found to be associated with the increased prevalence of MASLD. The MASLD prevalence trend was similar to that of NAFLD and steatotic liver, and as 96% of MASLD cases were NAFLD, the increase in the prevalence of MASLD was thought to be largely attributable to the increase in the prevalence of steatotic liver resulting from imbalanced fat intake, as previously reported [18]. In this study, we also referred to national data on exercise habits; however, there was no statistical change in the percentage of habitual exercisers from 2010 to 2019. Therefore, dietary nutrition seemed to play a major role in the increase in the prevalence of steatotic liver and MASLD in both males and females.

It is noteworthy that the prevalence of GMDs decreased despite an increase in the prevalence of MASLD in both males and females. The decrease in the prevalence of GMDs was due to a decrease in the prevalence of prediabetes in both sexes, whereas the prevalence of T2DM remained unchanged. However, the prevalence of DM in Japan slightly increased in males and decreased in females, from 12.6% to 13.8% in males and from 7.8% to 7.7% in females, in the 2009 and 2019 national surveys, respectively, as noted by Goto et al. [19]. In addition, the World Bank Report [20] showed that the prevalence of DM in Japan decreased from 7.7% in 2011 to 6.6% in 2021. Although our data were limited to one region and the majority (55.9–58.4%) of the total population throughout the study period was middle-aged, which differs from the age structure of the Japanese general population, we do not consider this result unique in terms of a decrease in the prevalence of prediabetes or GMDs. Liver steatosis and glucose intolerance are closely linked and are mutually promoting factors. Therefore, the increase in the prevalence of steatotic liver and decrease in the prevalence of GMDs are anomalous, and we focused on dietary changes in the Japanese population as the cause.

National data showed a slight increase in EN intake over the past decade for both males and females, but a decreasing trend in CHO intake and the CHO/EN ratio. According to the guidelines of Hauner et al. [21], there is insufficient evidence of an association between T2DM risk and CHO intake and no association with intake ratio, and, in a dose-response meta-analysis by Hosseini et al. [22], a CHO/EN ratio of 45–65% was not associated with T2DM risk. However, it is unknown whether prediabetes, and the possibility that this change contributed to the decrease in the prevalence of prediabetes in this study, can be ruled out.

On the other hand, dietary fat is increasing, and high-fat diets have been reported to induce insulin resistance and T2DM [23,24]. Furthermore, SFAs induce insulin resistance [25]. Conversely, high fat does not contribute to the risk of T2DM [26,27] and an inverse association between total lipids and T2DM or prediabetes has recently been reported [28,29]. In addition, certain FAs have been reported to improve or suppress GMDs [27,30,31,32]. In a multiple-treatment meta-regression, Imamura et al. [33] demonstrated that the conversion of CHOs to SFAs lowered fasting insulin levels, the conversion to MUFA lowered HbA1c and homeostasis model assessment of insulin resistance, and the conversion to PUFA improved these outcomes, in addition to FPG and C-peptides.

In a dose–response meta-analysis, Neuenschwander et al. [26] reported a dose-dependent effect of vegetable fat, SFAS, and n-6 FAs on T2DM risk reduction. In the NHNS, SFA, MUFA, and PUFA intakes in Japan showed an increasing trend, and this change may have been responsible for the reduction in the prevalence of prediabetes in this study. Lipids are also closely related to the development of steatotic liver, and it has been reported that SFAs directly induce hepatic steatosis [34,35], whereas MUFA and n-6 FAs are both reported to act in an inhibitory manner [36,37]. Therefore, it may be possible to control both hepatic steatosis and abnormal glucose metabolism by maintaining an appropriate balance between the intake of these lipids and EN.

Hypertension decreased and salt intake was weakly correlated with hypertension in males and moderately correlated in females. Although several causes of hypertension exist, salt intake is extremely important. Salt intake decreased more slowly in females, but was correlated with a decrease in the prevalence of hypertension in females. This may be largely due to the lower absolute salt intake in females.

Individuals with MASLD had a significantly higher frequency of overweight, HWC, hypertension, and T2DM than individuals without MASLD. MASLD and NAFLD are almost identical [38] and their clinical backgrounds are similar. In a recent large Japanese study by Fujii et al. [31], the rates of hypertension and dyslipidemia in patients with NAFLD were slightly lower than those in our results. This may be due to the fact that the diagnostic criteria for LRDs in this study were based on the International Consensus Group criteria, as well as regional characteristics. Regarding laboratory values, MASLD deviated more from the reference values than non-MASLD, which is consistent with the previous report [11] and may be owing to a higher prevalence of LRDs. The large number of LRD holders in the MASLD demonstrates the importance of the fact that the MASLD was created for the purpose of proactively capturing at-risk individuals.

The limitations of this study included the following. First, the results were from a specific region of Japan, and the nutritional data were from national data and not from the participants themselves. This was a retrospective study and no nutritional surveys were conducted; therefore, data were not available. However, this is a typical medium-sized Japanese city, 60 km from the capital, with an average annual income that is approximately equal to the Japanese average, although slightly lower than the Japanese average for primary industry workers. Furthermore, the patient population was comprised of ordinary people. Thus, as the nutrition data were averages for Japan, the correlations were not expected to diverge significantly. However, future research should be conducted over a wider area, including participants’ nutritional and physical activity status.

The second limitation was the diagnosis of LRDs. In this study, we diagnosed abnormal blood pressure as hypertension; abnormal lipid metabolism as dyslipidemia—based on the International Consensus Group criteria [6]; and abnormal glucose metabolism as T2DM and prediabetes combined—based on the American Diabetes Association criteria [13]. However, these are slightly different from the Japanese diagnostic criteria. Therefore, it is necessary to discuss the criteria that should be adopted when studying MASLD in Japan.

The third was that HOMA-IR was not included as a GMD diagnostic criterion. This may have resulted in a lower rate of GMD diagnosis. However, this impact was expected to be equally felt throughout the entire period, and the effect on the overall trend in prevalence was considered to be minor.

The fourth limitation was the inclusion of only physical examinations. Thus, no genetic or molecular data were available and no studies were conducted in this regard. Regarding genetic predisposition, many genetic polymorphisms have been reported to be involved in NAFLD and LRDs [39]. Among them, patatin-like phospholipase domain-containing 3 (*PNPLA3*) is thought to be involved in the onset and progression of NAFLD in Japan [40,41]. Recently, associations between fat intake and *PNPLA3* polymorphism and fish intake and transmembrane 6 superfamily member 2 (*TM6SF2*) polymorphism have been reported with respect to NAFLD development [42,43]. Interactions between LRD-related genes and various foods and nutrients have also been reported [44,45,46]. Future studies on the relationship between NAFLD and metabolic abnormalities or diet should include such genetic polymorphisms.

## 5. Conclusions

The prevalence of MASLD increased in all age and sex groups from 2010 to 2019, as did the prevalence of SLD itself, while the prevalence of prediabetes decreased. A change in diet, primarily lipid-based, was considered a contributing factor in both cases.

## Figures and Tables

**Figure 1 medicina-60-01330-f001:**
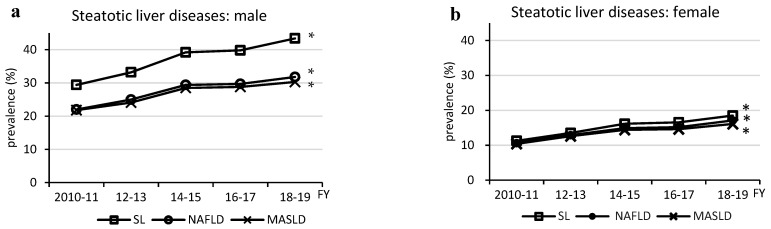
Changes in prevalence of steatotic liver diseases. FY, fiscal year; MASLD, metabolic dysfunction-associated steatotic liver disease; NAFLD, non-alcoholic fatty liver disease; SL, steatotic liver; SLD, steatotic liver disease; * (ρ = 1.0, *p* < 0.05).

**Figure 2 medicina-60-01330-f002:**
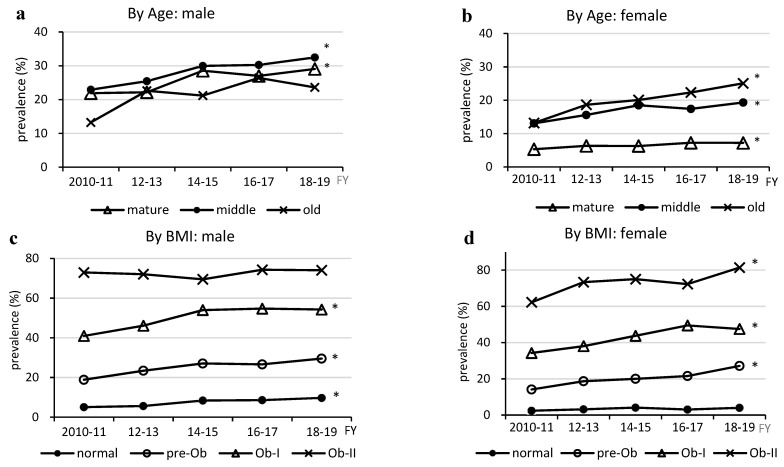
Changes in prevalence of MASLD by age and BMI. MASLD, metabolic dysfunction-associated steatotic liver disease; BMI, body mass index; FY, fiscal year, * *p* < 0.05.

**Figure 3 medicina-60-01330-f003:**
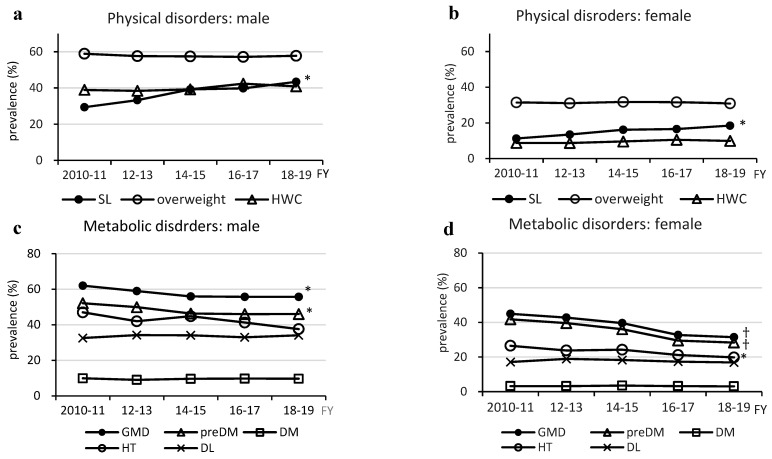
Changes in prevalence of metabolic dysfunction-associated steatotic liver disease-related physical and metabolic disorders. Physical Disorders (**a**,**b**). Metabolic Disorders (**c**,**d**). DL, dyslipidemia; DM, type 2 diabetes; FY, fiscal year; GMD, glucose metabolism disorder; HT, hypertension; HWC, high waist circumference; preDM, prediabetes; SL, steatotic liver * *p* < 0.05; † *p* < 0.005.

**Figure 4 medicina-60-01330-f004:**
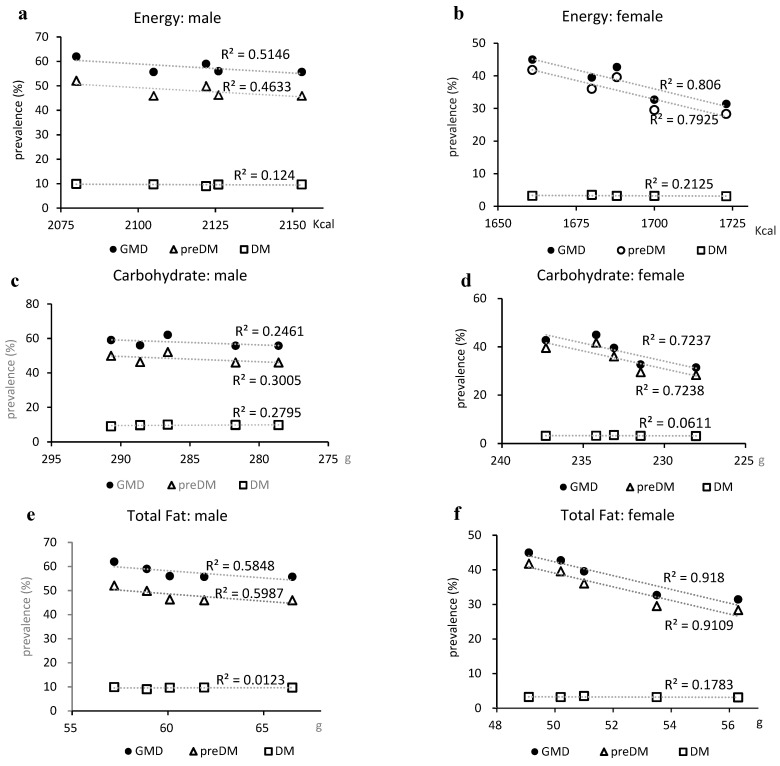
Relationship between glucose metabolism disorders and nutrient intake. (**a**,**b**) Total energy intake, (**c**,**d**) total carbohydrate intake, (**e**,**f**) total fat intake, (**g**,**h**) saturated fatty acid intake, (**i**,**j**) monounsaturated fatty acid intake, and (**k**,**l**) polyunsaturated fatty acids (sum of n-3 and n-6 polyunsaturated fatty acid). DM, type 2 diabetes; GMD, glucose metabolism disorder; MUFA, monounsaturated fatty acids; preDM, prediabetes; PUFA, polyunsaturated fatty acids; SFA, saturated fatty acids.

**Figure 5 medicina-60-01330-f005:**
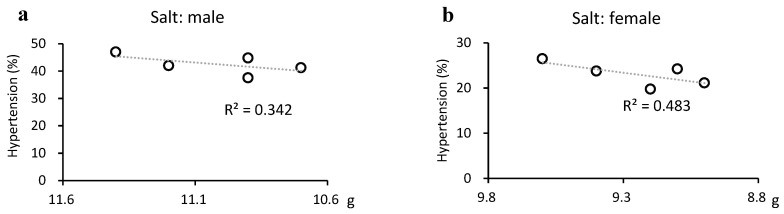
Relationship between hypertension and salt intake. Salt intake was significantly reduced in males (from 11.4 g to 10.9 g, *p* = 0.037) but weakly correlated with the prevalence of hypertension (R^2^ = 0.342). However, there was a moderate correlation with a reduced prevalence of hypertension in females (R^2^ = 0.483), where the reduction in salt intake was not significant (from 9.6 g to 9.2 g, *p* = 0.089).

**Table 1 medicina-60-01330-t001:** Characteristics of the subjects.

	2010–2011		2012–2013		2014–2015		2016–2017		2018–2019	
Number of subject	5358		5840		6062		6726		7392	
male (n)	2829		3102		3177		3425		3705	
female (n)	2529		2738		2885		3301		3687	
male										
Age category, n (%)										
25–44 years	964	(34.1)	1006	(32.4)	1048	(33.0)	1011	(29.5)	1129	(30.5)
45–64 years	1631	(57.7)	1773	(57.2)	1775	(55.9)	1967	(57.4)	2110	(57.0)
65–79 years	234	(8.3)	323	(10.4)	354	(11.1)	447	(13.1)	466	(12.6)
BMI status Category, n (%)										
<23 kg/m^2^	1163	(41.1)	1315	(42.4)	1352	(42.6)	1467	(42.8)	1565	(42.2)
23–24.9 kg/m^2^	743	(26.3)	820	(26.4)	802	(25.2)	876	(25.6)	935	(25.2)
25–29.9 kg/m^2^	801	(28.3)	831	(26.8)	869	(27.4)	903	(26.4)	997	(26.9)
≥30 kg/m^2^	122	(4.3)	136	(4.4)	154	(4.8)	179	(5.2)	208	(5.6)
female										
Age category, n (%)										
25–44 years	885	(35.0)	951	(34.7)	1004	(34.8)	1076	(32.6)	1177	(31.9)
45–64 years	1477	(58.4)	1567	(57.2)	1607	(55.7)	1889	(57.2)	2127	(57.7)
65–79 years	167	(6.6)	220	(8.0)	274	(9.5)	336	(10.2)	383	(10.4)
BMI status Category, n (%)										
<23 kg/m^2^	1732	(68.5)	1887	(68.9)	1968	(68.2)	2256	(68.3)	2545	(69.0)
23–24.9 kg/m^2^	383	(15.1)	391	(14.3)	430	(14.9)	469	(14.2)	494	(13.4)
25–29.9 kg/m^2^	324	(12.8)	355	(13.0)	375	(13.0)	457	(13.8)	503	(13.6)
≥30 kg/m^2^	90	(3.6)	105	(3.8)	112	(3.9)	119	(3.6)	145	(3.9)

BMI, body mass index.

**Table 2 medicina-60-01330-t002:** Trend of dietary nutrients and exercise habits (2010–2019).

		Male				Female		
	2010–2011	2018–2019	ρ	*p*	2010–2011	2018–2019	ρ	*p*
Energy (Kcal)	2080	2153	1	<0.0001	1661	1723	0.9	0.0374
Protein (g)	74.5	78.6	0.7	0.1881	62.3	66.2	1	<0.0001
Carbohydrate (g)	286.6	278.6	−0.7	0.1881	234.2	228.0	−0.4	0.5046
Carbohydrate/En %	61.3	57.8	−0.9515	0.0590	58.8	55.3	−0.9619	0.0499
Fat (g)	57.2	66.5	1	<0.0001	49.1	56.3	1	<0.0001
SFA (g)	14.9	19.0	1	<0.0001	13.2	16.6	1	<0.0001
MUFA (g)	19.8	24.9	1	<0.0001	16.6	20.9	1	<0.0001
PUFA (g)	12.7	14.4	0.9636	<0.0001	10.6	12.1	0.8061	0.0049
Salt (g)	11.4	10.9	−0.9	0.0374	9.6	9.2	−0.8208	0.0886

Nutrient intakes are listed only for the first and last fiscal year of the five periods from 2010–2011 to 2018–2019 that were analyzed by Spearman’s rank correlation test. Carbohydrate/En %: percentage of energy intake from carbohydrates, SFA: saturated fatty acid, MUFA: monounsaturated fatty acid, PUFA: polyunsaturated fatty acid (total of n-3 and n-6 PUFA).

**Table 3 medicina-60-01330-t003:** Comparison of MASLD and non-MASLD.

	Non-MASLD		MASLD		*p*-Value	MASLD %
Variables	N = 5.677		N = 1.715		MASLD vs. Non-MASLD	In Variable
Male	2.582	(45.5)	1.123	(65.5)	<0.0001	
Age ^†^	50.2	(10.2)	52.0	(9.6)	<0.0001	
Steatotic liver	575	(10.1)	1.715	(100.0)	<0.0001	74.9
Body mass index ≥ 23.0 kg/m^2^	1.820	(32.1)	1.462	(85.2)	<0.0001	44.5
High WC	835	(14.7)	1.044	(60.9)	<0.0001	55.6
Hypertension	1.357	(23.9)	764	(44.5)	<0.0001	36.0
Dyslipidemia	1.007	(17.7)	878	(51.2)	<0.0001	46.0
Glucose metabolism disorders	2.058	(36.3)	1.165	(67.9)	<0.0001	36.1
Type 2 diabetes mellitus	220	(3.9)	252	(14.7)	<0.0001	53.4
Prediabetes	1.838	(32.4)	913	(53.2)	<0.0001	33.2
Excessive alcohol intake	1.353	(23.8)	0	(0)		
Body mass index (kg/m^2^) ^†^	21.9	(3.0)	26.5	(3.8)	<0.0001	
*Laboratory test* ^†^						
Platelet (10^4^/µL)	25.0	(5.8)	26.2	(8.7)	<0.0001	
AST (U/L)	21.3	(8.8)	26.0	(13.1)	<0.0001	
ALT (U/L)	19.4	(14.0)	34.8	(25.7)	<0.0001	
GGT (U/L)	33.7	(43.4)	46.1	(45.1)	<0.0001	
Triglyceride (mg/dL)	89.7	(69.9)	142.8	(98.2)	<0.0001	
HDL-C (mg/dL)	67.8	(15.2)	54.0	(11.8)	<0.0001	
FPG (mg/dL)	97.9	(13.9)	108.1	(22.2)	<0.0001	
HbA1c (%)	5.4	(0.4)	5.8	(0.9)	<0.0001	

Values are expressed as number (%) or ^†^ mean (standard deviation). MASLD, metabolic dysfunction-associated steatotic liver disease; WC, waist circumference; AST, aspartate aminotransferase; ALT, alanine aminotransferase; GGT, gamma-glutamyl transferase; HDL-C, high-density lipoprotein cholesterol; FPG, fasting plasma glucose; HbA1c, hemoglobin A1c.

## Data Availability

The data presented in this study are not publicly available due to privacy and ethical restrictions.

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
