# Peer review of "Metabolic Dysfunction-Associated Steatotic Liver Disease in Japan: Prevalence Trends and Clinical Background in the 10 Years before the Coronavirus Disease 2019 Pandemic"

_medicina, 2024, doi:10.3390/medicina60081330_

Round 1

Reviewer 1 Report

Comments and Suggestions for Authors

I congratulate the authors for the work done. The manuscript brings many elements of novelty and is of interest for many medical specialties. However, to a preliminary read, I have some major concerns regarding the scientific work presented:

- MASLD is a concept that includes at least one metabolic disease in its definition and the afirmation that “The increase in the prevalence of MASLD was due to an increase in that of steatotic liver itself, and there was no increase in the prevalence of other factors, such as overweight, T2DM, hypertension, and dyslipidemia” which is found in the Abstract and further discussed in the manuscript is not legit, as the decrease in metabolic diseases prevalence should lead to a decrease in MASLD, and the increase in liver steatosis which was observed is due to other causes and therefore does not overlap MASLD definition. Furthermore, when discussing limitations of the manuscript, the authors recognize that the criteria for metabolic disease used in the paper are the ones for Caucasians, which “are slightly different from the Japanese diagnostic criteria”. In my opinion the analysis should be remade using ethic appropriate criteria.

- I find the assessment of the dietary habits and physical exercise in the Japanese population very interesting, but I suggest the authors to separate the two studies, as it is hard to assume that there is an overlap in these regards between the study population evaluated in the original study, over a decade, and the data that came from the populational study.

Comments on the Quality of English Language

The overall quality of English Language is good, requiring only minor editing.

Author Response

Dear, Reviewer

I am grateful for your excellent comments.

Please find the file attached.

Akira Sato

Reviewer 2 Report

Comments and Suggestions for Authors

In the present work (Medicina-3106202), Sato et al., have performed epidemiological analyses to find trends in MASLD in a Japanese population. Given the global epidemic of metabolic diseases including MASLD such studies are highly significant. The study is nicely conducted with a large cohort which is balanced in major criterions. The article is well written and I have only some minor suggestions/concerns, listed below.

Concerns:

1.       The introduction is very small and needs to be little more descriptive.

2.      In the introduction the authors need to provide references for “ numerous cross-sectional epidemiological studies” and also for “ few longitudinal epidemiological studies”.

3.      Did this study also evaluate HOMA-IR? This needs to be mentioned?

4.      For some figures it will be good to indicate p-values inside the panels for easy comparison.

5.      Really appreciate the limitations section in the discussion. The authors can also add a statement that an absence of GWAS and molecular study is also a limitation.

Author Response

Dear Reviewer

Thanks for your advice, it is much appreciated.

Please find our response in attached file.
